# How to learn more about hydrological conditions and phytoplankton dynamics and diversity in the eastern English Channel and the southern bight of the North Sea: the SRN data set (1992-2021).

Lefebvre Alain[1], Devreker David[1]

[1] Ifremer, Unité Littoral, Laboratoire Environnement et Ressources, 150 quai Gambetta, BP 699, 62321 Boulogne-sur-mer, France.

*Correspondence to*: Alain Lefebvre (alain.lefebvre@ifremer.fr)

**Abstract.** This article describes a thirty-year data series produced by the SRN ('Suivi Régional des Nutriments' in French; Regional Nutrients Monitoring Programme) network managed by Ifremer. Since 1992, the SRN network has been analysing

phytoplankton species and measuring physicochemical (temperature, salinity, oxygen, suspended matter, nutrients) and biological (chlorophyll-*a*, phytoplankton abundance) parameters at ten different stations distributed along three different transects located in the Eastern English Channel, and the Southern Bight of the North Sea. This geographic coverage allows for the study of three distinct ecosystems, i.e., three transects (estuary, coastal region under freshwater influence, and coastal region), as well as the investigation of coastal to offshore water gradients. The SRN collects a maximum of 184 samples per

year (3687 samples spread over 10 stations during the studied period) and detects up to 291 taxa, including harmful algal bloom species (HABs), with a bi-weekly to monthly sampling frequency (depending on the location and the season). The objectives of this monitoring program are to assess the influence of continental inputs on the marine environment, and their implications on possible eutrophication processes. It also aims to estimate the effectiveness of development and management policies in the marine coastal zone by providing information on trend and/or shifts in pressure, state, and impact variables. The

regular acquisition of data allows the establishment of long-term monitoring of the evolution of coastal water quality as well as the observation of the consequences of large-scale alterations mainly driven by climate change and modifications that are more related to local/regional anthropogenic activities. This paper provides an overview of the main characteristics of SRN data (descriptive statistics and data series main patterns) as well as an analysis of temporal trends and shifts. We also propose to the data user a specific numerical tool available as an R package to optimize the data pre-processing and processing steps.

Users will then have easy access to statistics, trends, and anomalies as proposed in this paper. The main results of several research projects based on SRN data and dealing with hydrology, phytoplankton blooms, HAB, phenology, and niche are also highlighted, providing the readers with examples of what is can be done with such a data set. We hope that this synthesis will also save data users time by allowing them to jump right into a deeper analysis based on previous conclusions and perspectives, or to investigate new scientific key challenges. These data should also be used at a wider geographical scale, combined with

other data sources, to define more global patterns of environmental changes in a moving world subject to strong anthropogenic

pressures. Data can also be used by the remote sensing (Ocean Color Observation) and modeling communities to calibrate or validate products in this complex and vital coastal region.

**Copyright statement**

**1 Introduction**

Phytoplankton contribute to the biological pump that regulates the flow of carbon dioxide. This compartment is critically important as it forms the basis of marine food webs. Understanding the structure of this community is essential for any assessment of marine diversity (Garmendia et al., 2013). Hence, maintaining ecosystem goods and services is partly linked to phytoplankton dynamics. There are several thousand phytoplankton species on the planet; the vast majority are completely

harmless, but a few hundred can proliferate significantly, forming red, brown, or green waters, a few dozen are toxic to marine fauna or humans (for example, through shellfish consumption (toxin bioaccumulation process)). Some can cause excessive organic matter inputs, well known as foam, which is produced, for example, by the Prymnesiophyceae *Phaeocystis globosa* and affects water quality (Lancelot et al., 1994). While some taxa contribute naturally to energy transfers to higher trophic levels, others are responsible for the development of toxic algal blooms that limit grazing (Nejstgaard et al., 2007). Indeed,

Harmful Algal Blooms (HAB) can reduce benthic and pelagic biodiversity, and disrupt marine ecosystems (Rousseau et al., 1990). They also may lead to the degradation of seafood quality, rendering it unsuitable for human consumption (Berdelet et al., 2016). As a result, any imbalance within the community, such as favoring dinoflagellates over diatoms, will have major effects on the biodiversity of higher trophic levels (Henson et al., 2021), as well as on the quality of the ecosystem in general (Lefebvre & Devreker, 2020). Consequently, such modifications of ecosystem goods and services will lead to several socio-

economic consequences.

The observation and monitoring of ecosystems quality is often accomplished by setting up networks to monitor hydrological and biological parameters, which constitute the essential foundation for the overall characterization of the aquatic ecosystem. As a result, the environmental descriptors are numerous, and are linked by cause and effect relationships; others are directly influenced by anthropogenic activities. Their ability to rapidly respond to changes in their environment allows the evaluation

of a response to different sources of pressure. Phytoplankton is regularly used as a water quality indicator for directives and conventions (e.g. the Water Framework Directive (EU WFD, 2000), the Marine Strategy Framework Directive (MSFD - 2008/56/EC) (European Commission MSFD, 2008) and the Oslo and Paris Convention (OSPAR Commission, 2013)). The metrics created for their purposes frequently employ phytoplankton biomass, abundance, and composition, as well as the frequency and intensity of blooms.

For a long time, the English Channel and the southern Bight of the North Sea have been subjected to intense anthropogenic pressure (fishing, tourism and leisure activities, marine aggregate extraction, maritime traffic, major port areas, degraded estuarine areas, off-shore wind turbine projects, etc.), with considerable economic stakes and subject to a diverse range of users with frequently antagonistic interests. The English Channel ecoregion is an epicontinental sea that serves as a transition zone between the Atlantic and North Sea water bodies. The region has a temperate oceanic climate, influenced by wet and

cold atmospheric currents coming from the Atlantic or more sporadically from the North Sea. It is characterized by its megatidal hydrodynamic regime, induced by tidal currents. This intense hydrodynamics influences the nature, distribution and dynamics of sediments, as well as the structure, distribution, dynamics and functioning of biological compartments. In addition, a maritime zone under the impact of freshwater (ROFI: Region Of Freshwater Influence) arises along the French shores of the eastern Channel, mostly from the Seine river, and whose structure is maintained by contributions from other northern rivers

(Somme, Canche, Authie). This structure will play an essential role in controlling the exchange of inert or living particles, organisms between the coast and the open sea (Brylinski et al., 1991).

    Phytoplankton data allow us to study harmful algal blooms (HABs), particularly *Phaeocystis globosa* and *Pseudo-nitzschia* ones. Since the 1970s, increase in intensity and duration of *Phaeocystis globosa* blooms in the North Sea have inspired scientific teams to launch research projects dedicated to this taxon in the context of excessive nutrient enrichment of marine

waters (Admiraal and Venekamp, 1986; Cadée and Hegeman, 1986; Eberlein et al., 1985; Lancelot and Mathot, 1985; Lancelot et al., 1997; Lefebvre et al., 2011; Lefebvre and Dezécache, 2020; Rahmel et al., 1995). *Phaeocystis globosa* was then identified as a potentially harmful taxon (in the HAB sense), not just due to its toxicity, but rather to the huge biomass created during its blooms and its consequences. *P. globosa* is also likely to produce precursors of dimethyl sulphide (DMS), a gas with a sulphurous and unpleasant odour. It was reported that DMS could cause respiratory, skin and ocular irritation in humans. It

can also be found in the atmosphere and can be favourable to the formation of acid rain. At the end of its very complex, polymorphic development cycle, *P. globosa* appears in colonial form. These colonies are loaded with mucco-polysaccharides. They will break up in response to internal factors (ageing, lysis) and/or external factors (turbulence) and provoke by emulsion the accumulation of a thick, odorous foam on the coast. The *Pseudo-nitzschia* complex contains species that can produce an amnesic phycotoxin based on domoic acid. During *P. globosa* blooms, needles formed by *Pseudo-nitzschia* complex cells can

stick into *P. globosa* colonies (Sazhin et al., 2007). We believe that such a structure may irritate filter feeders. The lesions caused by these structures may promote viral and bacterial infections in fish.

    Prior to 1992, French monitoring of phytoplankton populations and associated environmental factors in the English Channel and the southern bight of the North Sea was done episodically, via the RNO (Réseau National d'Observation) or the RNC (Réseau National de Contrôle). The Artois-Picardy Water Agency and Ifremer established the SRN (Suivi Régional des

Nutriments) in 1992 in response to the need for precise monitoring of nutrient concentration variations over a longer period, as well as the response of the environment to this pressure, particularly in terms of phytoplankton development. The objective of this monitoring is to evaluate the influence of continental inputs on the marine environment (nitrogen, phosphate, silicate) and their consequences on possible eutrophication processes. It also aims to estimate the efficiency of wastewater treatment

plants and policies for the development and management of the coastal zone and more generally the possible elimination of

such discharges. The regular acquisition of data allows the establishment of a long-term monitoring of the coastal waters along

three transects located off Dunkerque, Boulogne-sur-Mer and in the Bay of Somme, making it possible to pretend to be able

to deconvolute the effects of large-scale changes from those linked to more local anthropic activities (Lefebvre et al., 2011).

Since 1992, the SRN dataset has included long-term time series on marine phytoplankton and physico-chemical measurements

along the east coast of the English Channel and the French section of the southern bight of the North Sea (Dunkerque). SRN

data are complementary to REPHY and REPHYTOX datasets (PHYTOBS, 2021; REPHY, 2021; REPHYTOX, 2021).

Phytoplankton data cover microscopic taxonomic identifications and counts, up to the species level, and pigments measures

(Chlorophyll-*a* and pheopigment). Physico-chemical measurements include temperature, salinity, turbidity, suspended matters

(organic, mineral), dissolved oxygen, and dissolved inorganic nutrients (ammonium, nitrite+nitrate, phosphate, silicate).

## 2 Objectives

The objective of this paper is to present the SRN dataset, from the sampling strategy to data collection (including associated

Quality Assurance/ Quality Controls Steps) and data investigation and storage. The characteristics of the different datasets as

well as a general interpretation of their variability will be presented. Based on existing valorisations of the SRN dataset, we

will demonstrate that these data are of relevance not just for furthering understanding in marine phytoplankton ecology, but

also for public policy needs such as assessment of environmental or ecological status as requested by EU directives or Regional

Sea Conventions. We also present some numerical tools based on an R package available for the scientific community and

developed specifically to rapidly process such data and therefore to valorise the findings.

## 3 Materials and Methods

### 3.1 Sampling area and sampling stations

The English Channel has a macro-tidal regime that ranges from 3m to 9m in the Dover Strait during neap tide and spring tide,

respectively. This regime produces high tidal currents that are basically parallel to the shore, as well as a north-eastward tidal

residual current from the English Channel to the North Sea. Along the French coast, fluvial supplies dispersed from the Bay

of Seine to the Cape Gris-Nez form a coastal water mass that floats nearshore, protected from the open sea by a frontal region

(Brylinski et al., 1991). Exchanges between inshore and offshore water masses, as well as particle and nutrient transport, are

tide-dependent and are larger during the neap than during the spring tide. This may seem counter-intuitive, but during the neap

tide, the frontal structure between inshore and offshore waters is more sloped from the vertical resulting in a greater surface of

exchange between coastal and offshore waters. This leads to enhanced exchange possibilities between the two water masses

(Brylinski et al., 1991).

Three sampling areas along an inshore offshore transect were studied by Ifremer from 1992 to 2021 (still ongoing) within the frame of the SRN ("Suivi Régional des Nutriments", i.e. Regional Nutrient Monitoring Programme), two of which are located

in the Eastern English Channel and one in the southern bight of the North Sea (Figure 1):

1) Boulogne-sur-Mer harbour (by the Dover Strait), a coastal zone separated from the open sea by a frontal area (Station name: BL1, BL2, BL3);

2) Bay of Somme, the second ranked estuarine system after the Seine estuary on the French coasts of the English Channel (Station name: Bif, Mimer, Atso, Mer2);

3) Dunkerque harbour, a shallow well-mixed coastal zone near the frontier between France and Belgium (Station name: DK1, DK3, DK4).

The database shows another sampling area in the Bay of Somme, named "SRN Somme mer 1", but the survey was stopped in 2015. Therefore, these data will not be analysed in the present article, which is related to the 1992-2021 period.

The coordinates of sampling areas are given in the data base in columns "Coordonnées passage : Coordonnées maxx" for

longitude and "Coordonnées passage : Coordonnées maxy" for latitude (see supplementary materials) and summarized in table 1.

The positions of Mimer and Bif sampling stations have slightly changed as a function of the Bay of Somme sedimentary deposition (boat accessibility), figure 1 and table 1 present the most recent locations.

The main environmental characteristics of the areas are summarized in Table 2.


**Table 1. Coordinates of the different sampling areas of the SRN monitoring network.**

|  | Longitude | Latitude |
|---|---|---|
|  | WGS84 (decimal) | |
| Point 1 Dunkerque | 2.3334994588 | 51.0686501641 |
| Point 3 SRN Dunkerque | 2.2821638411 | 51.1089840828 |
| Point 4 SRN Dunkerque | 2.2508285996 | 51.1513182889 |
| Point 1 Boulogne | 1.5486581799 | 50.7531320816 |
| Point 2 SRN Boulogne | 1.5189908781 | 50.7531317402 |
| Point 3 SRN Boulogne | 1.4494895347 | 50.7481307969 |
| Atso | 1.4745046439 | 50.2312831163 |
| Mimer | 1.549383 | 50.235667 |
| Bif | 1.5986744158 | 50.214117386 |
| SRN Somme mer 2 | 1.441667 | 50.233333 |

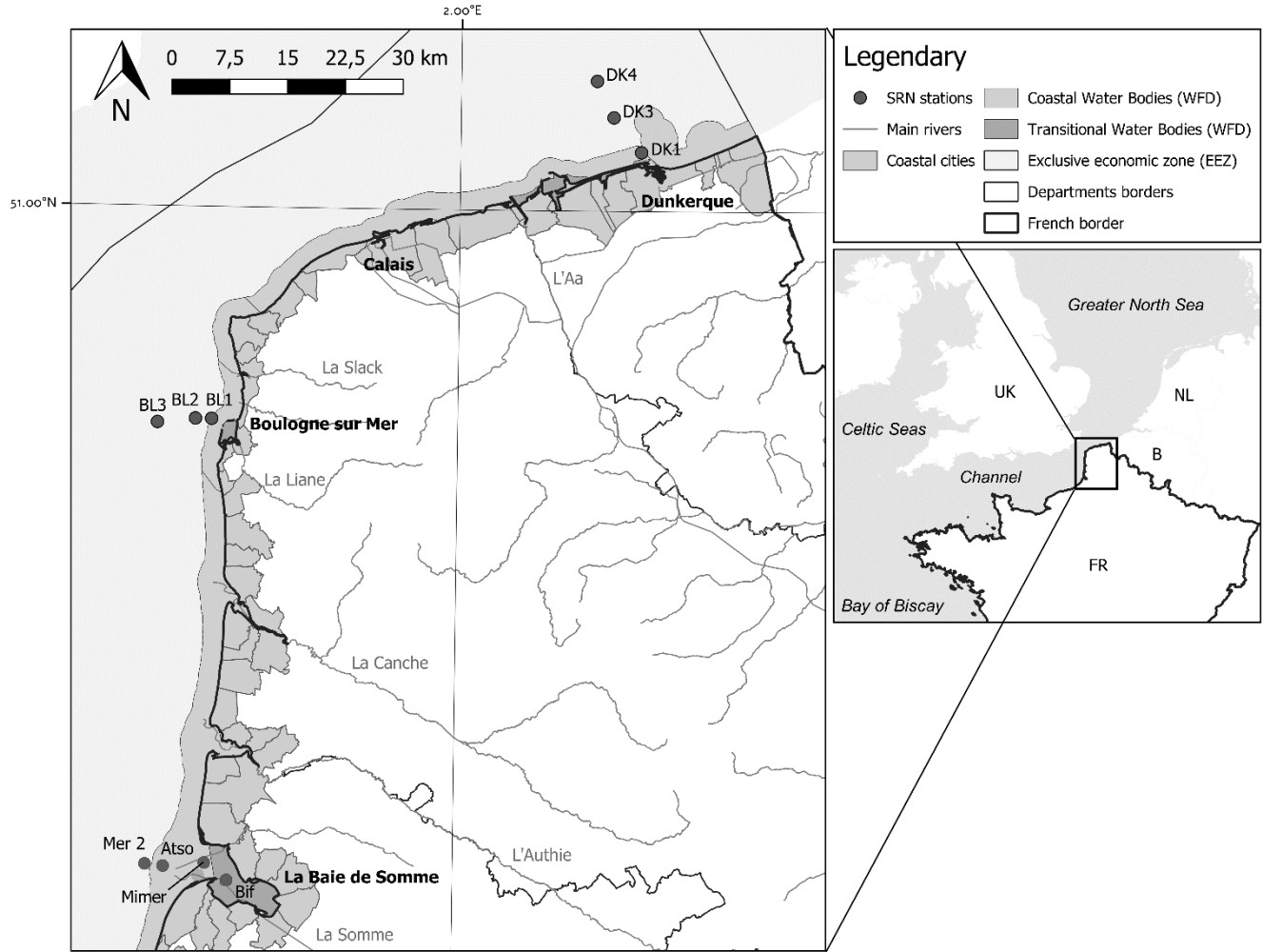

Figure 1. Map representing the different transects and sampling stations (Dunkerque, Boulogne and Bay of Somme) of the SRN network. The boundaries of the coastal and transitional water bodies of the European Water Framework Directive and the French Economic Exclusive Zone are also indicated.

Table 2. Main environmental characteristics of the three ecosystems/transects: Dunkerque, Boulogne-sur-Mer and the Bay of Somme.

|  | **Dunkerque** | **Boulogne-sur-Mer** | **Bay of Somme** |
|---|---|---|---|
| Tidal regime | macrotidal | macrotidal | macrotidal |
| Main characteristic | Shallow water (< 30 m) | Under the influence of a frontal structure | Estuary with medium turbidity |
| Main River | Aa | Liane | Somme |

| Main pressures | Industrial activities | Agricultural activities | Agricultural activities, shellfish aquaculture |
|---|---|---|---|
| Length of the main river (km) | 89 | 37 | 263 |
| Watershed (km$^2$) | 1215 | 244 | 6550 |
| Mean river Flow (m$^3$ s$^{-1}$) | 10 | 3 | 35 |
| Sampling stations | Point 1 Dunkerque (DK1) Point 3 SRN Dunkerque (DK3) Point 4 SRN Dunkerque (DK4) | Point 1 Boulogne (BL1) Point 2 SRN Boulogne (BL2) Point 3 SRN Boulogne (BL3) | Atso Mimer Bif SRN Somme mer 2 (Mer2) |

## 3.2. Hydrology

From March to June, water samples were taken from the coastal stations twice a month from subsurface waters using a 5 L Niskin Bottle, and once a month the rest of the year. For the intermediate and offshore stations, a monthly sampling strategy is implemented. Of course, this strategy is subject to adaptation when considering available human resources and/or
meteorological conditions, leading to a total number of samples lower than 184 for some years.

Based on water samples, the approach used for chlorophyll-*a*, ammonia, nitrite, nitrate, phosphate and silicate analyses is provided by Aminot & Kérouel (2004). Chlorophyll-*a* concentrations were estimated by spectrophotometry after filtration through glass fiber filters and extraction with 90 % acetone.

An accurate test of nutrient limitation requires detailed measurements of algal growth under experimental nutrient addition
(D'Elia et al., 1986). Nevertheless, in order to determine the potential limitation of primary production by nutrient availability when such experiment results are not available, the standard molar ratio for dissolved inorganic nitrogen (DIN = ammonium + nitrite + nitrate), phosphate and silicate were calculated and compared, according to the references of Redfield et al. (1963) and Brzezinski (1985) for the composition of the biogenic matter (Si: N: P = 16: 16: 1).

Salinity, temperature and oxygen concentration are measured at the same frequency (twice a month between March and June
and once during the rest of the year) during probe casts. For coastal stations, surface and bottom values (~1 meter above the seabed) are kept in the dataset. They are used to define bottom oxygen concentrations as requested in the WFD and MSFD for indicator calculation. For offshore stations, only surface values are considered. Our paper focuses only on surface values, but all the data are available from the DOI.

## 3.3. Phytoplankton

Phytoplankton samples have been collected along transects and were preserved with an acid lugol solution (0.25%). Sub-samples of 10 ml were settled for 24 hours in a counting chamber according to the Utermöhl (1958). Cell enumerations were performed by inverted microscopy using a microscope within a month after the sample collection to prevent any significant changes in phytoplankton size and abundance. Except for *Phaeocystis globosa* enumeration, over 400 phytoplankton cells in each sample were counted with a 20X Plan Ph1 0.5NA objective, resulting in an error of 10%. This species is reported as the genus *Phaeocystis* in the database due to national standardization. Nevertheless, for our area, Rousseau et al. (2013) confirmed that *P. globosa* is the relevant species. For assessment of *P. globosa* counts, only the total number of cells is computed. A minimum of 50 solitary cells were enumerated from several randomly chosen fields (10 to 30) with a 40X Plan Ph2 0.75NA. Abundance of cells in colony was determined using a relationship between colony biovolume and cell number defined by Rousseau et al. (1990).

Some harmful algae are also subject to specific surveys during a given risky period. Therefore, those species are part of a predefined list that must be checked during enumeration. If any of these taxa are absent from a sample, their abundance is reported as 0 in the database.

Phytoplancton identification is standardized using the WoRMS (2022) database and reaches the level of the species in many cases. However, when the identification is not easy or is subject to caution, a lower taxonomic level is kept in the Quadrige² database. Some species are also regrouped in "artificial taxa" if they are subject to strong identification confusion from analysts (this is the case for *Pseudo-nitzschia* or *Chaetoceros*, for example).

Species richness calculations are also based on all taxonomic levels.

A further detailed description of the REPHY procedure for phytoplankton identification and enumeration is proposed in Belin et al. (2017). The SRN strategy is fundamentally the same.

The total number of sample collected for complete determination of phytoplankton community since the creation of the SRN is shown in table 3.

**Table 3. Number of samples of phytoplankton (including replicates by date), number of dates of survey and number of taxonomic groups (species, genera or higher) by station between 1992 and 2021 at the different SRN stations.**

|  | Nb of dates | Nb of samples | Nb of toxonomic groups |
|---|---|---|---|
| DK1 | 392 | 393 | 221 |
| DK3 | 324 | 324 | 202 |
| DK4 | 309 | 309 | 191 |
| BL1 | 481 | 485 | 215 |

| | | | |
|---|---|---|---|
| BL2 | 400 | 400 | 208 |
| BL3 | 394 | 394 | 188 |
| Bif | 414 | 414 | 206 |
| Mimer | 272 | 272 | 197 |
| Atso | 468 | 471 | 224 |
| Mer2 | 390 | 390 | 197 |
| **Total** | **3844** | **3852** | **291** |

## 4. Database

To manage coastal monitoring data, Ifremer has developed the Quadrige² information system (https://envlit.ifremer.fr/Quadrige-la-base-de-donnees; last access on 24 January 2023), which combines a database with a
variety of interpretation and information product development tools. As an information system, Quadrige² plays a crucial role in: (1) storing basic monitoring data, such as the results of analyses from all monitoring networks, in a safe, optimal, supervised, and scalable manner, and (2) interpreting and valuing the data. Once the data has been stored and a quality level assigned to it, it is ready for use in a wide range of applications. As a result, this system is the required link for monitoring data between data collection in the field and its availability in various formats. Quadrige² has been approved as the national reference
information system for coastal waters by the French Ministry in charge of the Environment.

The datasets presented in this article are extractions from the Quadrige² database. The data come in two different files: one with hydro-chemical parameters and chlorophyll-*a* concentrations, containing 57 columns and 54 578 rows, and another with phytoplankton abundance data, containing 55 columns and 98 904 rows. These data are available as a csv file with a semi-colon separator. They are ANSI encoded and points are used as decimal separators.

Moreover, the database header's column are in French language (Quadrige² is a French national database). The French-English translation is given in the supplementary materials [S1]. In addition, a complete description of the header is given in Ifremer (2017).

Physico-chemical and phytoplankton data are being given and made available over the 1992-2021 period because the database is up-to-date until 2021 at the time the article is written. Because the database is updated annually, future users are likely to
download a dataset with additional years of data than the actual one. Former datasets (associated with the oldest DOI) are still available on user demand.

The phytoplankton database contains three different kinds of cell enumeration: FLORTOT, FLORPAR and FLORIND. In the column "Résultat : Code parameter", FLORTOT means that all taxa contained in the samples were identified, FLORPAR means that only some dominant taxa were identified when abundance > 100.000 cells per liter, and FLORIND means that only

indicative HAB taxa (*Dinophysi*s sp., *Alexandrium sp.* and *Pseudo-nitzschia* complex) were identified. For species richness indices and phytoplankton community distribution, only FLORTOT results were considered.

## 5. Quality Control

### 5.1. Data Validation

The data are collected in the field and/or laboratory, then enter into the Quadrige² database using an application with the same
name. The control entails modifying the data entered (results and metadata) to ensure that it is consistent with the bench book (or field sheets). After this check is completed and any necessary corrections are made, the data are validated:

- Confirmation of the technical validity of the data which corresponds to the result of the analysis,

- Data locking, so that it can no longer be edited, even by the person who entered it,

- Data distribution: verified data may be taken and disseminated by all Quadrige² users who have access to the database.

### 230 5.2. Data Qualification

This initial round of data verification is followed by the qualifying procedure, which corresponds to:

- the search for questionable or even scientifically aberrant data,

- the correction of data when possible,

- the attribution of a qualification level to the data, which is:

- good: data analysis are scientifically relevant,

    - doubtful: the data may be false: taking it into account may bias the analysis that will be made,

    - False: the data should not be included in the analysis because they are aberrant or present a problem (e.g., bad analytical series and impossible to repeat).

The level of qualification corresponds to the level of confidence in the data. It determines the way in which the data is
distributed (only data qualified as "good" and "doubtful" are distributed), and how it is used in specific data processing. The qualification is broken down into two main steps:

1) An "automatic" qualification that consists of looking for "gross" and easily identifiable errors,

2) An "expert" qualification, which consists in highlighting statistically aberrant data via adapted methods (time series analysis, statistical tests...). Only data qualified as "good" or "doubtful" from the previous step are used for the expert qualification.

## 245 6. Data Summary

Table 4 represents the descriptive statistics obtained for each physicochemical and biological parameter (excluding phytoplankton) and for each SRN station. For each of these series, the monotonic trend is estimated using a non-parametric method, Mann-Kendall Seasonally adjusted autocorrelated series test (Devreker and Lefebvre, 2014).

**Table 4. Statistical summary (minimum, first and third quantiles, mean, median, maximum, length of the data series) for the physico-chemical and biological variables collected within the SRN monitoring programme on the time span 1992-2021 and for coastal stations only (DK1, BL1, Atso). Increasing or decreasing monotonous[1] / non-monotonous[2] trends are indicated in the Sign. Trend column (1. One orange (green) arrow for an increasing (decreasing) monotonous trend - 2. Two arrows for a non-monotonous trend (shift in the time series) – Grey arrow indicates no significant trend).**

| Stations | Parameters | Min | 1st Qu | Median | Mean | 3rd Qu | Max | N | Sign. Trend |
|---|---|---|---|---|---|---|---|---|---|
| DK1 | Temperature (°C) | 1.00 | 8.30 | 12.10 | 12.35 | 16.60 | 21.70 | 404 | → (grey) |
| | Chlorophylle-*a* (µg.l⁻¹) | 0.240 | 2.500 | 4.490 | 6.642 | 7.790 | 53.180 | 421 | → (grey) |
| | Pheopigment (µg.l⁻¹) | 0.040 | 0.900 | 1.545 | 2.166 | 2.545 | 27.660 | 418 | ↗ (orange) |
| | Salinity | 31.10 | 33.70 | 34.21 | 34.07 | 34.60 | 35.50 | 424 | → (grey) |
| | SM (mg.l⁻¹) | 1.60 | 5.70 | 9.70 | 13.17 | 16.70 | 95.20 | 399 | → (grey) |
| | NO3+NO2 (µmol.l⁻¹) | 0.110 | 0.600 | 2.320 | 8.024 | 13.990 | 54.770 | 410 | ↘ (green) |
| | NH4 (µmol.l⁻¹) | 0.110 | 0.420 | 1.670 | 2.418 | 3.500 | 29.400 | 413 | ↘ (green) |
| | Oxygen (mg.l⁻¹) | 6.910 | 8.050 | 8.810 | 9.206 | 9.880 | 19.200 | 225 | ↘ (green) |
| | PO4 (µmol.l⁻¹) | 0.0100 | 0.1600 | 0.3800 | 0.5015 | 0.6500 | 9.8000 | 411 | ↘ (green) |
| | SIOH (µmol.l⁻¹) | 0.100 | 1.070 | 3.165 | 4.938 | 6.577 | 35.200 | 412 | ↗↘ (orange/green) |
| BL1 | Temperature (°C) | 2.10 | 8.80 | 12.50 | 12.64 | 16.70 | 22.10 | 484 | → (grey) |
| | Chlorophylle-*a* (µg.l⁻¹) | 0.010 | 1.730 | 3.380 | 5.082 | 7.025 | 29.600 | 495 | ↗↘ (orange/green) |
| | Pheopigment (µg.l⁻¹) | 0.040 | 0.940 | 1.610 | 2.339 | 2.874 | 14.900 | 490 | ↘ (green) |
| | Salinity | 29.14 | 33.60 | 34.10 | 33.93 | 34.50 | 35.30 | 492 | ↗ (orange) |
| | SM (mg.l⁻¹) | 0.050 | 3.425 | 5.600 | 8.208 | 9.350 | 46.400 | 470 | → (grey) |
| | NO3+NO2 (µmol.l⁻¹) | 0.150 | 0.600 | 2.000 | 6.605 | 11.300 | 43.810 | 474 | → (grey) |
| | NH4 (µmol.l⁻¹) | 0.0600 | 0.4475 | 1.0400 | 1.4355 | 1.9500 | 10.2000 | 476 | ↘ (green) |
| | Oxygen (mg.l⁻¹) | 6.260 | 7.940 | 8.640 | 8.912 | 9.800 | 12.500 | 249 | → (grey) |
| | PO4 (µmol.l⁻¹) | 0.0500 | 0.1300 | 0.2700 | 0.3927 | 0.5300 | 3.1000 | 478 | ↘ (green) |
| | SIOH (µmol.l⁻¹) | 0.100 | 0.500 | 1.800 | 3.193 | 4.312 | 19.010 | 476 | ↗ (orange) |
| At so | Temperature (°C) | 2.000 | 8.925 | 13.000 | 12.994 | 17.350 | 22.100 | 454 | → (grey) |
| | Chlorophylle-*a* (µg.l⁻¹) | 0.210 | 2.990 | 5.770 | 8.204 | 10.650 | 58.530 | 456 | → (grey) |
| | Pheopigment (µg.l⁻¹) | 0.000 | 1.617 | 3.050 | 6.099 | 6.272 | 66.070 | 452 | ↘ (green) |
| | Salinity | 26.00 | 32.20 | 33.20 | 32.76 | 33.70 | 35.10 | 461 | ↗ (orange) |
| | SM (mg.l⁻¹) | 0.90 | 7.00 | 13.48 | 19.35 | 25.77 | 167.00 | 451 | ↘ (green) |
| | NO3+NO2 (µmol.l⁻¹) | 0.10 | 1.70 | 6.85 | 11.99 | 19.39 | 56.86 | 446 | → (grey) |
| | NH4 (µmol.l⁻¹) | 0.020 | 0.365 | 0.890 | 1.942 | 2.430 | 30.700 | 447 | ↘ (green) |
| | Oxygen (mg.l⁻¹) | 4.060 | 8.200 | 9.010 | 9.239 | 9.980 | 13.980 | 237 | → (grey) |

| | | | | | | | |
|---|---|---|---|---|---|---|---|
| PO4(µmol.l⁻¹) | 0.0300 | 0.1350 | 0.2700 | 0.4135 | 0.5950 | 3.0300 | 447 |
| SIOH (µmol.l⁻¹) | 0.060 | 1.100 | 3.505 | 6.096 | 10.012 | 41.000 | 446 |

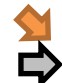


The seasonal variability of phytoplankton populations corresponds to maximum abundance in spring and then a decrease in winter. This trend can be variable depending on the sites (different hydrodynamical conditions) and the environmental characteristics encountered (luminosity, nutrient inputs, etc.). The water masses sampled appear to be poorly structured vertically, while significant coastal to offshore gradients are evident. Most of the time, the water masses are homogeneous.

The few vertical temperature variations are low, even at the coast, where the mixing of fresh and salt water is likely to create stratification. This is, in any case, negligible compared to the horizontal heterogeneity. The possibility of vertical stratification is more likely in the Baie de Somme. Thus, the general dynamics of the Eastern Channel-Southern Bight of the North Sea ecosystem represent the classical functioning of a temperate system (Wafar et al., 1983; Gentilhomme and Lizon, 1998). The seasonal cycles of nutrients and phytoplankton biomass are well-defined. Inter-annual variability is high. The homogeneity of

the sampling conditions makes it possible to avoid normalizing the results of nutrient concentration by salinity for the purpose of inter-site comparison. The analysis of the results shows a monotonic decreasing trend of phosphate concentration for all the studied sites, while silicate concentrations are relatively stable (except when considering inter-annual variability). The dynamics of nitrogen and phytoplankton biomass are far more complicated to handle, and such monotonic trends are not identifiable. These results may alter values of the stoichiometric ratios N: P: Si: N and Si: P (N: total nitrogen; P: Phosphate;

Si: silicate) (Redfield et al., 1963; Brzezinski, 1985). Phytoplankton growth appears to be primarily limited by P and Si (for diatoms only). This notion of limiting phytoplankton growth and its consequences on phytoplankton communities deserves particular attention in a system bordered by coastal marine regions where eutrophication problems are of great importance (Bay of Seine and the North Sea) (Lefebvre and Devreker, 2020). Indeed, considerable changes in phytoplankton productivity, abundance, dominance, and succession have been observed in recent decades as a result of increased human stresses,

particularly, nutrient inputs (Billen et al., 2005; Gypens et al., 2013).

For a particular site, phytoplankton counts shows high seasonal, interannual, and spatial variability (Figures 2, 3). Maximum abundances are measured between March and June, mainly due to the presence of *Phaeocystis globosa*, which dominates the phytoplankton population. Diatoms and dinoflagellates make up the majority of the community over the rest of the year. Lower diversity indices also characterized the spring period (April, May) since *P. globosa* dominates the community (Figure 4).

Between March and June, the prymnesiophyceae *Phaeocystis globosa* is sampled on a regular basis at all sites, and its concentration can exceed one million cells per liter (Figure 2). During the rest of the year, some isolated cells may be detected. The Bay of Somme area had the highest concentration from 1992 to 2007, with more than $48.10^6$ cells per litre at its peak. Likewise, maximum concentrations on the Dunkerque and Boulogne-sur-Mer transects are high, but to a lesser extent, reaching over $29.10^6$ and $28.10^6$ cells per litre, respectively.

The genera *Alexandrium*, *Dinophysis* and *Pseudo-nitzschia*, which are potentially responsible for the production of PSP (Paralytic Shellfish Poison), DSP (Diarrheic Shellfish Poison) and ASP (Amnesic Shellfish Poison) toxins, respectively, are regularly observed from the water samples at the monitoring sites. It is worth noting that, even when the cell densities of these toxic genera exceed the alert thresholds, toxin analysis of shellfish collected from the same area concerned by this bloom can be surprisingly negative. In the investigated regions, toxicity seems to be only a potential that is not expressed, maybe because

of unfavorable environmental conditions.

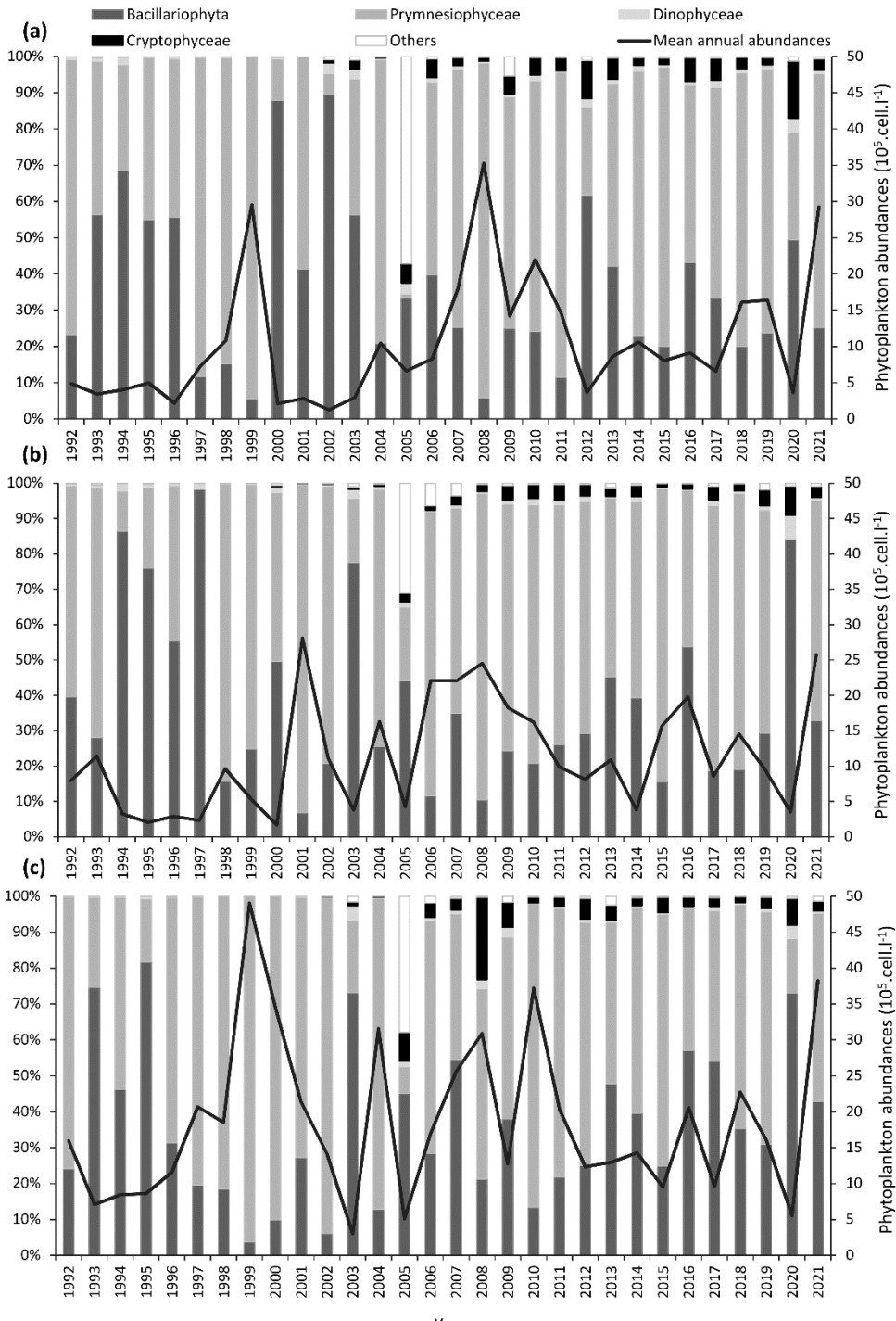

**Figure 2. Inter-annual variability of phytoplankton major groups (Prymnesiophyceae, Bacillariophyceae and others phytoplankton) at the three coastal SRN stations: (a) Dunkerque (DK1), (b) Boulogne-sur-Mer (BL1) and (c) Atso in the bay of Somme. Vertical bars represents the relative abundances of these groups (%) and the black line the mean annual total abundance ($10^5$ cell.L$^{-1}$).**

Figure 5 depicts the seasonal variability of the data. Despite the fact that the series shows drastically varied values depending on the station, the inter-seasonal variability for each of the characteristics depicted remains constant. In fact, nutrient concentrations are at their highest in winter. Phytoplankton that consume these nutrients increase during the spring, which explains why chlorophyll-*a* concentrations (a proxy for phytoplankton biomass) peak reaching its annual maximum, while oxygen concentrations decline reaching their annual minimum. At the end of summer, chlorophyll-*a* concentrations begin to diminish, while nutrients begin to replenish, eventually reaching high quantities in the fall. On the other hand, the temperature shows a classical variability of temperate marine waters with however some extreme values close to 0°C and above 20°C.

Figure 6 shows the calculated monthly-scale anomalies for temperature, chlorophyll-*a* concentration, and nutrients at each SRN coastal station.

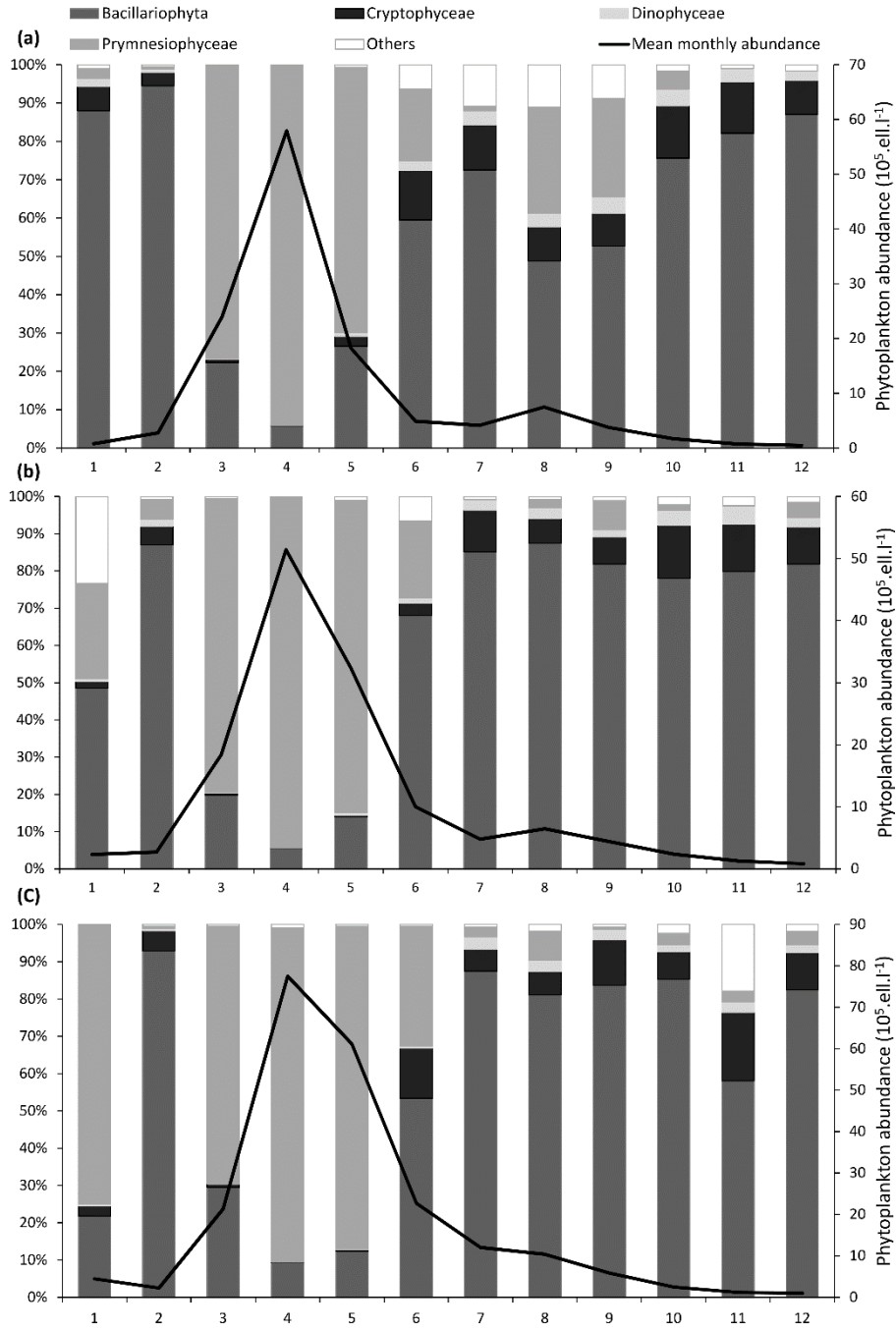

**Figure 3. Seasonal variability of phytoplankton major groups (Prymnesiophyceae, Bacillariophyta, Dinophyceae, Cryptophyceae and others phytoplankton) at the three coastal SRN stations: (a) Dunkerque (DK1), (b) Boulogne-sur-Mer (BL1) and (c) Atso. Vertical bars represents the relative abundance of these groups and the black line the mean monthly total abundance.**

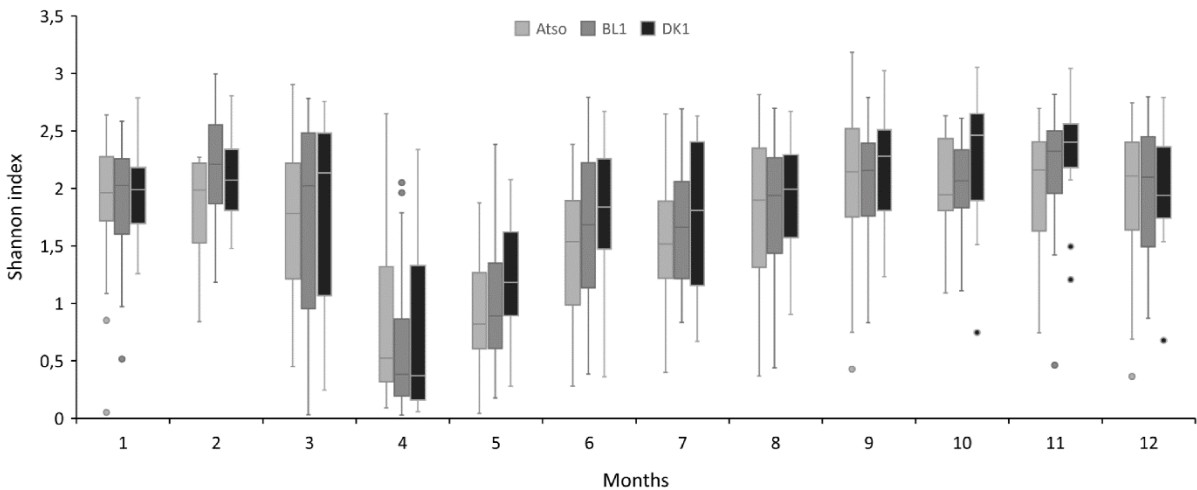

**Figure 4. Mean seasonal variation (monthly scale) of the Shannon entropy index at the different coastal stations (Atso, BL1, DK1).**


**Figure 5. Monthly box and whisker plots of main physico-chemical parameters for the three coastal stations (DK1, BL1 and Atso) of the SRN network for the period 1992-2021.**


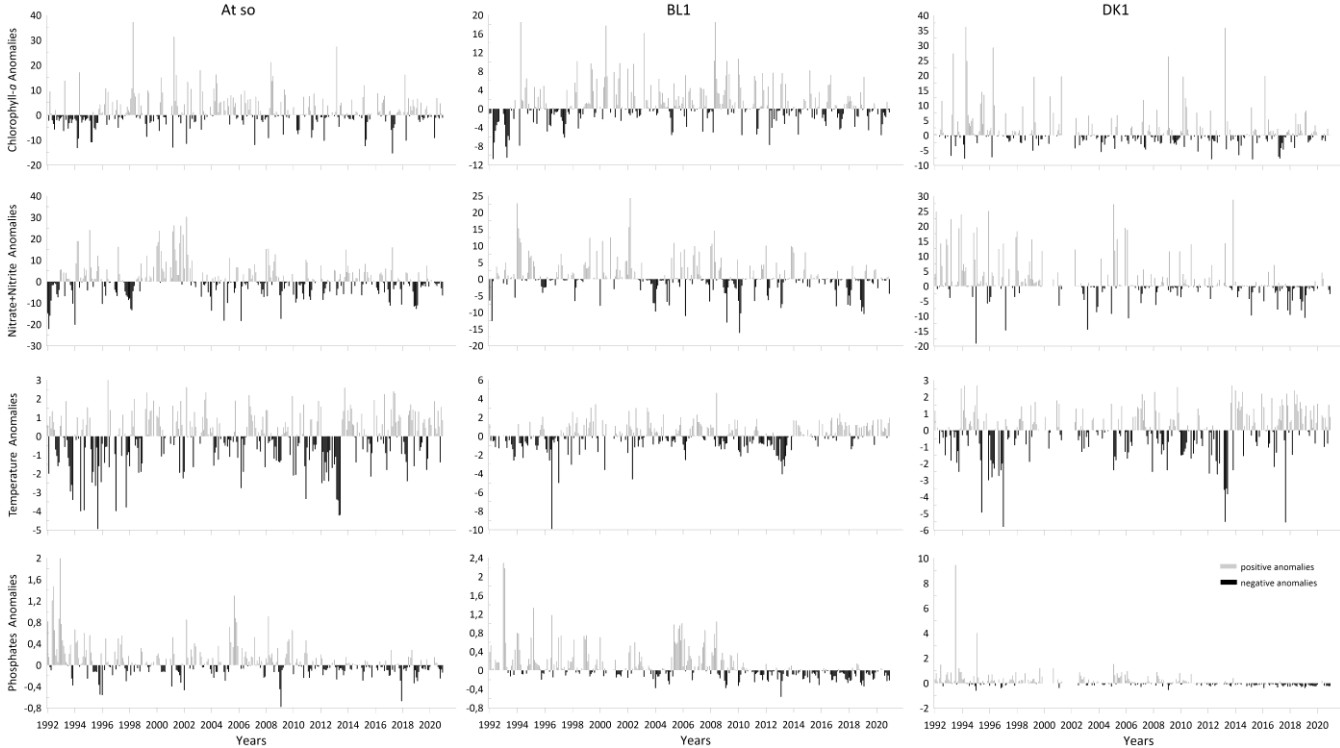

**Figure 6. Monthly anomalies calculated for Chlorophyll-*a* and nitrite+nitrate concentrations, temperature and phosphate concentration measured for the three coastal stations (DK1, BL1 and Atso) of the SRN network for the period 1992-2021.**

## 7. Discussion and conclusion


The SRN data, which have been collected since 1992, represent one of the longest long-term datasets in the English Channel and the southern bight of the North Sea. It enables the study of phytoplankton dynamics and diversity, as well as changes in particular composition in response to anthropogenic and/or climatic change. Indeed, this data have been used by scientists from several backgrounds, and for a variety of research purposes. In the following paragraphs, main outputs from these research

activities will be highlighted. This review can help the scientific community interested in this SRN dataset to identify topics that can be addressed or those that have not been addressed. It also allows for new topics to be discussed in more detail since the general patterns have already been identified.

Lefebvre et al. (2011) proposed a break and trend analysis for nutrient concentrations using a simple and intuitive analytical method called the cumulative sum method (Ibanez et al., 1993). The authors defined the characteristics and patterns of seasonal variations in chlorophyll-*a*, nutrients and phytoplankton. They proposed a classification of years based on whether *Phaeocystis globosa* is dominant or not.

According to Lefebvre et al. (2011) and Hernández-Fariñas et al. (2014), the phytoplankton community in the Eastern English Channel and Southern Bight of the North Sea is dominated by Bacillariophyta, Dinoflagellates, and Prymnesiophyceae, accounting for 81% of the total abundance. Hernández-Fariñas et al. (2014) estimated that the median contribution of *Phaeocystis globosa* during the period between March and May (data 1992-2014) ranges from 74% to 90% with the highest concentrations found at the coast. They highlighted two main periods with different environmental characteristics regardless of the SRN transect considered: 1992-2001 and 2002-2011. The latter period can be divided into two sub-periods: 2002-2007 and 2008-2011. These results highlighted the existence of a strong temporal structuring of the community under the influence of global and local factors. Globally, the abundance of *Pseudo-nitzschia* increases during the studied periods, while the abundance of other Bacillariophyta such as *Guinardia*, *Coscinodiscus - Stellarima* decreases. The dinoflagellates *Amphidinium*, *Alexandrium*, and *Polykrikos* mark the second great period. During the second sub-period, however, *Heterocapsa*, *Torodinium*, and *Eutreptiella* (Euglenoid) are widespread. *Melosira* and *Stephanopyxis* were frequent diatoms in the early years of monitoring, but became rare after 2002. There are changes in the abundance of some taxonomic units. The abundance of *Phaeocystis globosa* has not changed significantly in the Bay of Somme, although it has increased slightly in Dunkerque and Boulogne-sur-Mer. Between 2002 and 2007, the abundance of the *Gymnodinium-Gyrodinium* group of dinoflagellates increased significantly, corresponding to a log scale abundance increase of twofold.

More recently, Lefebvre & Dezécache (2020) highlighted a significant break in the evolution of *Phaeocystis globosa* and *Pseudo-nitzschia* complex abundance in the 2000s and different trajectories of abundances in response to changes in nutrient concentration observed over the period 1994-2018. The three contrasting SRN sites appear to respond differently depending on the intensity of the initial nutrient input pressure. While a recovery to a good ecological status is doubtful in the near future, these ecosystems appear to be in an unstable intermediate state that necessitates continuous efforts to reduce nutrient inputs, particularly nitrogen.

These considerations on the relationship between phytoplankton succession and environmental conditions inevitably lead us back to Margalef's (1978) mandala, who paved the way for phytoplankton ecology by proposing functional groups to represent the adaptation of different life forms to specific habitats. However, it seems very difficult to propose, based on these concepts, a typical pattern of phytoplankton succession in the eastern English Channel and southern Bight of the North Sea from the SRN data. Similarly, assigning phytoplankton succession to the classical path or the route leading to the harmful blooms of the Margalef's mandala is hard, as is proposing a logical transition scheme between the various strategies described. In fact, alternative routes, overlaps, and mixtures of taxa with distinct strategies do exist, and the same taxon can even display several strategies according to its morphotype, as in the case of *Phaeocystis globosa*.

The concept of niche is crucial in phytoplankton research because it helps in understanding the succession of taxa, their coexistence, exclusion, and the environmental conditions that control them, as well as their tolerance to environmental changes. Thus, Karasiewicz et al. (2018) used an improved version of the OMI (Outlying Mean Index) approach to evaluate the niches of diatoms and *Phaeocystis globosa*. Two different situations of *P. globosa* bloom amplitude were defined by two different environmental trajectories and two different diatom communities, whose key features are given in table 5. Karasiewicz & Lefebvre (2022) also developed a new method for bloom detection (based on twenty-two phenological variables) within a time-series. A pairwise quantification of asymmetric dependencies between the phenological variables revealed the implication of different mechanism, common and distinct between the studied taxa. A Permanova assisted in revealing the significance of seasonal variation in environmental and community factors. They were able to locate the harmful taxonomic niches among the rest of the community and quantify how their respective phenology influences the dynamic of their subniches by using methodologies such as the Outlying Mean Index and the Within Outlying Mean Index.

These results are comparable to those of Hernández Fariñas et al. (2015) who, based on a similar approach but extended to the Dunkerque and Bay of Somme SRN sites, concluded that light, temperature, species richness and nutrient concentrations are the main factors controlling phytoplankton dynamics and community structure.

**Table 5. Main biotic and abiotic characteristics during two contrasting situations of *Phaeocystis globosa* bloom intensity from SRN data in the coastal zone off Boulogne-sur-Mer (-: low value for the parameter under consideration; +: high value).**

| Bloom intensity for *P. globosa* | Low | High |
|---|---|---|
| Initiation of the *P. globosa* bloom | Late | Early |
| Temperature | - | + |
| Salinity | - | + |
| Turbidity | + | - |
| Winter nitrate stock | + | - |
| Winter phosphate stock | + | - |
| Competition with Diatoms | + | - |

We also confirmed from SRN data that periods of *Phaeocystis globosa* dominance are generally associated with high concentrations of *Pseudo-nitzschia* complex. The simultaneous presence of *P. globosa* and *Pseudo-nitzschia* complex will cause the creation of structures resembling mini-bearings (*Pseudo-nitzschia* needles planted in *Phaeocystis* colonies) that will be passively swallowed by the fish. These structures can cause mechanical aggression of the gill and/or digestive tissues opening the way to viral and bacterial infections. This mechanical aggressiveness will become even more crucial as new needle-shaped species or those with pointed spicules have been found in our investigations, contributing to the creation of these irritating assemblages. These are *Chaetoceros sp.*, some *Thalassiosira sp.*, *Rhizosolenia imbricata*, *R. styliformis*. Furthermore, the impact of phytoplankton blooms on the pelagic compartment, and in particular on fish, needs further

investigation, especially in light of the findings of Lefebvre et al. (2011), Hernández-Fariñas et al. (2014), who found an increase in the abundance of *Pseudo-nitzschia sp.* since the early 2000s, and Delegrange et al. (2018), who found a correlation between mortalities of farmed sea bass (*Dicentrarchus labrax*) and the spring phytoplankton bloom. Major blooms of *P.*
*globosa* will lead to changes in viscosity (Seuront et al., 2006) that may cause behavioural changes in fish (e.g. inhibition of larval swimming activity) or metabolic changes (e.g. inhibition of gill functions). Changes in viscosity will also inevitably affect prey/predator relationships within planktons (Seuront and Vincent, 2008).

In order to facilitate the analysis and valorisation of SRN data, Devreker & Lefebvre (2014) proposed the development of a
user interface developed in R (R Core Team, 2020). This interface is available on the Comprehensive R Archive Network (CRAN) as the TTAinterfaceTrendAnalysis package. It allows quickly defining the main statistical characteristics of SRN series and proposing classical time series analyses (data regularization and aggregation, detection of anomalies, breaks and seasonal or global trends). The results are presented in the form of summary tables or graphs that are automatically saved in the user's working directory.


SRN data are used for validation of coupled hydrodynamic-biogeochemical models such as ECO-MARS 3D, in addition to improving scientific knowledge about phytoplankton dynamics, biodiversity, and water quality (Ménesguen et al., 2019). Including retrieved-chlorophyll-*a* and suspended matters, SRN data are an essential source for the development and improvement of satellite water color algorithms (Gohin et al., 2019, 2020a). In terms of the latter parameter, they are the only
data with such geographical and temporal coverage available in the coastal area of the Eastern Channel and the southern bight of North Sea. Other data from the Coastal Observation Service (Somlit; https://www.somlit.fr/; last access on 28 April 2022) can supplement the SRN data for the Boulogne sur Mer coastal zone (Lheureux et al., 2021).
In the context of the implementation of the Water Framework Directive (WFD 2000/60/EC) since 2007 (COM, 2005, a, b, c), some coastal points of SRN integrate the so-called Monitoring and Operational Control system. The new Marine Strategy
Framework Directive (MSFD) extends the WFD approach limited to the first nautical mile from the baseline (for biological parameters) to the offshore waters (Exclusive Economic Zone). Thus, the offshore stations of SRN network also meet the diagnostic and monitoring expectations advocated by this European directive (Lefebvre and Devreker, 2020).
As part of the Oslo and Paris Convention's strategy to combat eutrophication (OSPAR http://www.ospar.org/), SRN results are used to define the eutrophication status of water bodies. The SRN data are also transmitted to the ICES working group
"Phytoplankton and Microbiol Ecology" (WG PME) in order to contribute to the drafting of the devoted annual report (http://www.ices.dk/community/groups/Pages/WGPME.aspx; last access on 24 January 2023).

**Data availability**

SRN - Regional Observation and Monitoring program for Phytoplankton and Hydrology in the eastern English Channel (2022). SRN dataset - Regional Observation and Monitoring Program for Phytoplankton and Hydrology in the eastern English
Channel. SEANOE. https://doi.org/10.17882/50832

**Related datasets:**

Coastal stations of the SRN regional networks are part of the REPHY national network. These networks share the same protocol and consider the same parameters. The PHYTOBS dataset (PHYTOBS is a SNO, National System of Observation, labeled network from the Research Infrastructure "Littoral et Côtière" ILICO) is part of the REPHY and SOMLIT (Observation
Services in Littoral Environment) network stations. The REPHYTOX is complementary to the REPHY network, as it contains phycotoxin concentrations in mussels from areas where HAB phytoplankton blooms are detected.

PHYTOBS (2021). PHYTOBS dataset - French National Service of Observation for Phytoplankton in coastal waters. SEANOE. https://doi.org/10.17882/85178
REPHY – French Observation and Monitoring program for Phytoplankton and Hydrology in coastal waters (2021). **REPHY dataset - French Observation and Monitoring program for Phytoplankton and Hydrology in coastal waters. Metropolitan data**. SEANOE. https://doi.org/10.17882/47248
REPHYTOX - French Monitoring program for Phycotoxins in marine organisms (2021). **REPHYTOX dataset. French Monitoring program for Phycotoxins in marine organisms. Data since 1987**. SEANOE. https://doi.org/10.17882/47251

**Code availability**

R package *TTAinterfaceTrendAnalysis* on the CRAN website (Comprehensive R Archive Network - https://cran.r-project.org/web/packages/TTAinterfaceTrendAnalysis/index.html ; last access on 24 January 2023).

**Team list**

Lefebvre Alain, Devreker David, Blondel Camille, Duquesne Vincent, Hebert Pascale, Fabien Lebon, Cordier Remy, Belin
Catherine, Maud Lemoine, Nadine Neaud-Masson, Huguet Antoine, Durand Gaétane, Soudant Dominique.

**Author contribution**

AL Alain Lefebvre led the SRN monitoring programme and also led the writing of the paper. DD David Devreker collaborated to the writing of the paper. Colleagues from the team list led either cruises or complementary monitoring networks (linked to

related datasets), either analysed the samples, or contributed to Quality Assurance and Quality Control procedures described in the paper.

## Competing interests

The authors declare that they have no conflict of interest.

## Acknowledgments

The acquisition of this dataset was performed in the framework of the SRN network, created and implemented by Ifremer and the Artois Picardie Water Agency in 1992. The people cited in the team list and contributors include regional coordinators of the SRN, key analysts and experts in phytoplankton observations, key analysts in physico-chemical measurements, those who contributed to structuring SRN data in the Quadrige² database, and individuals who contributed to the data through their personal expertise. Other contributors such as the REPHY and REPHYTOX teams, crewmembers, students and PhD who have contributed to valorisation of the SRN data are also gratefully acknowledged.

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
