# Peer review of "How to learn more about hydrological conditions and phytoplankton dynamics and diversity in the eastern English Channel and the southern bight of the North Sea: the SRN data set (1992-2021)."

_Earth System Science Data, 2022_

## Author Comment (AC1)

The authors propose a synthesis of data contained in databases over nearly 30 years. The interest is obvious, but the globalisation of the parameters leads to a loss of sight, and ends up pushing open doors. The authors need to go into much more detail with corelations (more original, finer and more precise... to be found) for this kind of article to be useful and also to correspond to the title (which I find particularly well selling but misleading).

In detail, not wanting to repeat what the two first referees have already pointed out:

- Line 30: not all toxicity phenomena for humans are through shellfish consumption,

*Text has been modified. L. 40-44.*

- Line 34/35: "...major effects on the biodiversity of higher trophic levels". Need a solid reference to back this up

*A reference from Nature Communication has been added in this sentence L.49. to illustrate this higher trophic level effect. We would also like to add a reference about water quality assessment, specifically eutrophication assessment within the EU Marine Water Framework Directive: this reference highlights direct and indirect effects of river nutrient inputs on phytoplankton biomass and at the ecosystem level (Lefebvre & Devreker, 2020)*

- Line 60: "...abnormal increase...", "...naturally occurring...". I think these considerations are no longer in the way of thinking... and without getting into philosophical debates!

*"Abnormal" and "naturally" have been removed and the sentence was modified. L.74-75.*

- Line 71/72: *Pseudo-N* needles stick into *Phaeocystis* colonies irritate filter feeders. Is this proven? Do they irritate more or less than in the isolated planktonic state?

*As far as we know, there is no specific reference paper about this potential effect. We have modified the sentence accordingly.*

*L.86-88. "We believe that such a structure may irritate filter feeders. The lesions caused by these structures may promote viral and bacterial infections in fish (AL, pers. comm.). »*

- Line 89: Phaeopigment. They are not used afterwards.

*This variable is now available from the DOI. The statistical summary was added to Table 4.*

- Line 193/196: These seem to me to be generalities that deserve to be detailed or referenced otherwise they do not belong here.

*We are not sure we understand this comment. This part of the manuscript is just a summary of the phytoplankton dynamics in our temperate ecosystems. We built our manuscript to allow the readers to find (i) a general description of the data and the main patterns in the present manuscript and (ii)*

*more details from the review of scientific works using SRN data.*

- Line 208: What is "Phytoplantonic taxonomic productivity"?

*This was an error. "taxonomic" has been removed.*

- Line 213: *P. globosa* is not a *prymnesiophyceae* but a *coccolithophyceae* in the current systematics. (idem in the legend of fig. 2)

*According to the Worms, P. globosa is a Prymnesiophyceae (Class) (https://www.marinespecies.org/aphia.php?p=taxdetails&id=160538)*

[Figure]

*But according to Algaebase, P. globosa is associated with the Coccolithophyceae class (https://www.algaebase.org/search/species/detail/?species_id=52922)*

*As a result, because Worms is our reference (Quadrige2 database), we chose Prymnesiophyceae.*

[Figure]

[Figure]

*About Bacilariophyceae, indeed, this is a class according to Worms, but some taxa (belonging to Diatoms\*) are not included in this Bacilariophyceae class (\*Central and et Pennal diatoms, phylum Ochrophyta).*

*According to Algaebase, the Bacillariophyta phylum contains only diatoms.*

*Consequently, we decide to use Bacillariophyta.*

- Lines 218-220: Potentially toxic but no toxin detected. OK but isn't that a bit short on explanation. Expliquer prélvt eau + coquillages + seuils

*The sentence has been modified to further explain this issue.*

*L. 293-298. "The genera Alexandrium, Dinophysis and Pseudo-nitzschia, which are potentially responsible for the production of PSP (Paralytic Shellfish Poison), DSP (Diarrheic Shellfish Poison) and ASP (Amnesic Shellfish Poison) toxins, respectively, are regularly observed from the water samples at the monitoring sites. It is worth noting that, even when the cell densities of these toxic genera exceed the alert thresholds, toxin analysis of shellfish collected from the same area concerned by this bloom can be surprisingly negative. In the investigated regions, toxicity seems to be only a potential that is not expressed, maybe because of unfavorable environmental conditions."*

- Fig 2: Only *bacillariophyceae* are taken into account? Why are not all diatoms considered?

*Figures 2 and 3 have been updated to include Bacillariophyta, Cryptophyceae, Dinophyceae and Prymnesiophyceae.*

- Line 259: Why use the term dinoflagellates when other algal groupings use taxonomic ranks?

*We used the same term as in the cited manuscripts.*

Line 266: The 3 diatoms mentioned are not *bacillariophyceae*. Idem for the following lines, there is a mishmash of terms.

*Guardinia, Stellarima and Coscinodiscus are bacillariophyceae (class) according to WORMS. However we change this to Bacillariophyta. L.346, 353.*

Line 386: Carpentier, Martin & Vaz: This is grey literature.

*We have deleted this reference. No such synthesis is available from other kinds of references.*

---

## Author Comment (AC2)

- The authors present a dataset from the shallow coastal waters in the eastern English Channel and southern bight of the North Sea collected from 1992-2001, primarily for the purpose of water quality monitoring, The data are reported to include 281 taxa and 3687 samples spread over 10 stations.

- Readers would greatly benefit from a careful and thorough editing of the text, in particular the abstract and introduction. At present ideas are sometimes out of order or disjointed and the purpose of the manuscript is not clear.

*The abstract and summary have been redesigned to better highlight the content and the research opportunities made possible by the availability of such a dataset.*

- The methods and results are quite clear. The overviews of the physical-chemical variables are very helpful. I was surprised to see no indication of dinoflagellates in Fig. 2.  Is it possible that there were no phytoplankton dinoflagellates (autotrophs or mixotrophs)? This seems unlikely. I wonder if a seasonal version of the count distribution would be helpful.  Are any size data or approximate biomasses of the enumerated taxa available? It might also be interesting to know something of the distribution of species richness, even just the average richness per sample.

*Dinoflagellates are identified in SRN samples but in lower abundance than Diatoms, Cryptophyceae, or Prymnesiophyceae, that is why we did not initially include them in the figure. Because it appears that this information might be of interest, we have added Dinophyceae and Cryptophyceae to figures 2, 3.*

*Cell size and carbon biomass are not measured in the SRN network, so there is no such information in the database.*

*We have added a figure with monthly mean variations of the Shannon index among coastal stations (figure 4).*

- I thank the authors for their efforts to disseminate and describe their wonderful dataset. Unfortunately, from my perspective the data are inadequately described. Since this is a data paper in a data journal, this is a problem.

*We have added more information in the 'abstract' and in the 'Materials and methods' sections.*

- I have examined the data at https://doi.org/10.17882/50832
- The data appear to be encoded in latin1 as a csv file with a semi-colon separator. It only takes a few guesses to work this out, but ideally the reader would not need to guess.

*Details about file format have been added in 4. Database section:*

*L. 214-215. « These data are available as a csv file with a semi-colon separator. They are ANSI encoded and points are used as decimal separators.. »*

- I did not see a data dictionary or other description of the contents of the file anywhere. The variable headings are written in French. I understand the desire to work in one's preferred language, but the language of publication is English, so a translation should be provided.

*Added in "4. Database" section :*

*L. 216-218. « Moreover, the database header's column are in French language (Quadrige² is a French national database). The French-English translation is given in the supplementary materials [S1]. In addition, a complete description of the header is given in Ifremer (2017).»*

- I was able to read a data table of 61 variables and 99,006 observations. The number of observations and variables should be reported in the metadata so that the user can be sure the data were received as expected.

*This information have been added in the 4. Database section:*

*L. 212-214. « The datasets presented in this article are extractions from the Quadrige² database. The data come in two different files: one with hydro-chemical parameters and chlorophyll-a concentrations, containing 57 columns and 54 578 rows, and another with phytoplankton abundance data, containing 55 columns and 98 904 rows. »*

- There are 6 variables which are missing for all observations. I don't understand the need to include undescribed variables with no data.

*During the extraction process, some empty columns appear automatically when other (non-empty) columns are selected. They have to be manually removed. This is done, and the datasets can now be downloaded without these empty columns.*

- There were some (809) zero counts for taxonomic abundance, but very few (<1%). Please clarify the reason for including these zeros. Was a consistent taxonomic list used for all stations and times? Can the reader infer that the taxa recorded at some stations but not others have zero abundance when not reported?

*Supplementary information about 0 data has been added in the 3.3 Phytoplankton section:*

*L. 184-186. "Some harmful algae are also subject to specific surveys during a given risky period. Therefore, those species are part of a predefined list that must be checked during enumeration. If any of these taxa are absent from a sample, their abundance is reported as 0 in the database."*

- It would be helpful to provide latitude and longitude of the stations; these can be read approximately from Fig. 1, but they do not appear to be in the data file. I was unable to decode the station location data: cordonnees passage min and max for x and y.

*A table with stations' coordinates was added in the article (table 1). Coordinates in the database are in decimal and can be found under "cordonnees passage minx and maxx" for longitude and "coordonnees passage miny and maxy" for latitude. For column header translation, see the supplementary materials S1 and the section 3.1. Sampling area and sampling stations:*

*L. 136-138. « The coordinates of sampling areas are given in the data base in columns "Coordonnées passage : Coordonnées maxx" for longitude and "Coordonnées passage : Coordonnées maxy" for latitude (see supplementary materials) and summarized in table 1. »*

- The values I computed for Table 2 did not match the authors', likely because I misinterpreted something; incomplete description of the data makes this easy to do unfortunately. I suggest additional details to clarify the differences.

*The calculations were redone, and new values appear in table 3 (formerly table 2) and in the related text.*

- Is there a problem with station SRN Somme mer 1 (Mer1?) resulting in it not being reported in Table 2?

*Additional information is given in part 3.1 Sampling area and sampling stations about this station:*

*L. 134-135. « The database shows another sampling area in the Bay of Somme, named "SRN Somme mer 1", but the survey was stopped in 2015. Therefore, these data will not be analysed in the present article, which is related to the 1992-2021 period. »*

- Number of observations (samples) per station:
- z1 |> count(lieu_de_surveillance_libelle, passage_identifiant_interne, passage_date) |> count(lieu_de_surveillance_libelle)
- # A tibble: $11 \times 2$
-  lieu_de_surveillance_libelle     n
-
-  1 At so                479
-  2 Bif              414
-  3 Mimer                272
-  4 Point 1 Boulogne         508
-  5 Point 1 Dunkerque          406
-  6 Point 2 SRN Boulogne       401
-  7 Point 3 SRN Boulogne       395
-  8 Point 3 SRN Dunkerque       324
-  9 Point 4 SRN Dunkerque       309
- 10 SRN Somme mer 1        301
- 11 SRN Somme mer 2        391
-
- The total number of samples in the dataset is 4200. This does not match Table 2 (3687) even if all the observations from SRN Somme mer 1 are removed. My calculations showed that 2007 had the most observations (179), but this does not agree with the number in the abstract (184, line 12). It would be more representative to report the mean or median number of observations (142, 140) or the range (100 to 179).

*184 is the theoretical number of samples that are scheduled by year, but due to meteorological problems (or COVID problems in 2020), some stations cannot always be sampled. The text was updated consequently at L. 155-158.*

*Data processing was repeated (without Mer1), and new values appear in table 2 (now table 3); the number of samples is 3 852.*

- Number of species:
- z1 |> count(resultat_nom_du_taxon_referent, lieu_de_surveillance_libelle) |> count(lieu_de_surveillance_libelle)
- # A tibble: $11 \times 2$
-    lieu_de_surveillance_libelle     n
-
-   1 At so                       224
-   2 Bif                         206
-   3 Mimer                        197
-   4 Point 1 Boulogne              215
-   5 Point 1 Dunkerque            221
-   6 Point 2 SRN Boulogne          208
-   7 Point 3 SRN Boulogne          188
-   8 Point 3 SRN Dunkerque         202
-   9 Point 4 SRN Dunkerque         191
-  10 SRN Somme mer 1              167
-  11 SRN Somme mer 2              197
- Has the taxonomy be standardized in any way? Was a database such as marinespecies.org used? I suspect the count of species in table 2 in fact refers to some level of taxonomic resolution and not species. In addition to species, the data table reports many genera, some higher classifications, and size or shape features. Some taxa are fusions of several species or genera, e.g., "Chaetoceros densus + eibenii + borealis + castracanei". Some classifications can be guessed, but are incomplete, e.g., "Centriques", "Pennées".

*Additional information is added in section 3.3. Phytoplankton:*

*L. 184-191. "Phytoplancton identification is standardized using the WoRMS (2022) database and reaches the level of the species in many cases. However, when the identification is not easy or is subject to caution, a lower taxonomic level is kept in the Quadrige² database. Some species are also regrouped in "artificial taxa" if they are subject to strong identification confusion from analysts (this is the case for Pseudo-nitzschia or Chaetoceros, for example). Species richness calculations are also based on all taxonomic levels."*

- I did not see any spelling errors in the taxonomic identifications; I congratulate the data curators for this success!

*Thank you!*

- Phaeocystis globosa was repeatedly identified in the manuscript, but does not appear in the dataset. Only the genus-level id Phaeocystis is reported in the data. This is a serious oversight or inconsistency. It would be helpful to note in the data if the counts refer to cells, colonies, or a mixture.

*Additional information is added in section 3.3. Phytoplankton :*

*L. 178-180. « This species is reported as the genus Phaeocystis in the database due to national standardization. Nevertheless, for our area, Rousseau et al. (2013) confirmed that P. globosa is the relevant species. »*

- I was able to read the physical-chemical data and interpret it. The general concerns above copy over here about documentation, encoding, location information. I did not see any information about the units of measurement of the various quantities (temperature, chl-a, nitrate, nitrite, phosphate, silicate; some can be guessed, but the nutrients could easily be in mass or mol units and there is no way to tell.) Guessing is not ideal in a documented dataset. The metadata indicated phaeopigments, suspended matter organic and mineral are reported, but I did not see any observations of these quantities in the data. These are oversights that should be corrected.

*Units are present in the dataset in the "Résultat : Symbole unité de mesure associé au quintuplet" and "Résultat : Libellé unité de mesure associé au quintuplet" columns. See the new supplementary materials S1 for translation and more details.*

*Pheopigments (PHEO) and suspended matters (MES and MESORG) should be present but were not extracted (mistakenly!). They are now present in the dataset.*

- I did not examine the data at the following sites as they did not seem to be the primary target of this data paper:
- https://doi.org/10.17882/85178 , https://doi.org/10.17882/47248 , https://doi.org/10.17882/47251.

*They are similar datasets (same database, same protocol, but not the same location and networks).*

- A bit more information at line 347 about the relationship between these data would be helpful. (Are they completely distinct, partially overlapping, etc.?)

*Added in the "Related dataset" section:*

*L. 435-439. «Coastal stations of the SRN regional networks are part of the REPHY national network. These networks share the same protocol and consider the same parameters. The PHYTOBS dataset (PHYTOBS is a SNO, National System of Observation, labeled network from the Research Infrastructure "Littoral et Côtière" ILICO) is part of the REPHY and SOMLIT (Observation Services in Littoral Environment) network stations. The REPHYTOX is complementary to the REPHY network, as it contains phycotoxin concentrations in mussels from areas where HAB phytoplankton blooms are detected.»*

- I was unable to use the R package TTAinterfaceTrendAnalysis. It requires X windows and Tk application software which many users will not have installed. It might be helpful to indicate something of the software requirements in a brief note. These requirements are a bit unusual for modern software packages and will likely limit the usage of their package.

*The package should run since the official CRAN quality control was implemented during package submission. The package can work with any operating system and automatically install the dependent packages. Tcl-tk is part of the basic installation of R.*

- Detailed comments
  Abstract
  The abstract does not clearly describe the dataset, which is the main purpose of this paper. I suggest informing the reader of the years covered by the data and the total number of observations.

*The abstract was updated.*

- The abstract is written about an "historical" dataset, yet is largely written in the present tense (SRN collects, objectives … are, regular acquisition of data…) This is a bit confusing, indeed the paper both describes an ongoing program and presents data from 30 years of collection. A bit of smoothing of the exposition and clarification of the goals of the manuscript would help the reader.

*The abstract was updated.*

- Line 8. Define acronym SRN before it is used (It is defined repeatedly below)

*Done. ('Suivi Régional des Nutriments' in French; Regional Nutrients Monitoring Programme) added in the abstract from the first citation of the SRN acronym.*

- Line 8. Give location of Ifremer (Brest)

*Ifremer is an institute distributed over several cities and regions in France, so not all laboratories are located in Brest. This precision is not useful in the abstract as we identified the studied area in the title and gave the author's address.*

Line 8. What makes the data historical? Has data collection ceased? Are modern data excluded?

*We have modified the abstract. The SRN network is still active. The DOI is updated each year with the integration of new data after QA/QC controls. Each data set identified by a DOI at a given time remains accessible on demand.*

- Line 12. 184 samples in one year would be quite intense sampling. Is this phrasing correct? Samples per station and the number of stations might be more informative.

*More details about the number of transects, the number of stations, and the related number of samples are given in the updated abstract.*

- Line 14. Are continental inputs, development and management policy metadata part of the time series? Are these data described in this manuscript? Available publicly?

*No, they are not part of the time series*

- Introduction
- Line 30: "others cause excessive organic matter inputs". I think I know what you mean here, but this is a relatively unusual observation, so an example taxon or citation could help make your point more clearly.

*Done. L. 43.*

- Line 59. What does "address" mean here? It's a fairly indirect verb.

*"Address" was changed to "study"*

- Line 74. Presumably French should be capitalized here.

*Done. L. 70.*

- Purpose

*?*

- Line 97. Should "propose" be "present"?

*Done. L. 112.*

---

## Author Comment (AC3)

This is an interesting paper, describing a set of measurements conducted by the SRN network in the Eastern English Channel, and the Southern Bight of the North Sea.

The dataset described in this paper consists of measurements of hydrological and ecological variables, going back to 1992. Such historical datasets contribute to our understanding of long term ecological processes, and improve our ability to monitor and preserve sensitive ecosystems. This is especially important in areas subject to strong anthropogenic pressure, as the one addressed here, making the described dataset interesting and valuable.

The paper is well organized and clear. The authors do well in describing the geographical, hydrological and ecological context, emphasizing the datset's importance and relevance. The technical aspects of data collection, analysis methods and quality control are, in general, well-described, although some important corrections/clarifications should be made (see comments below). In addition, the authors provide useful description of observed trends, discuss possible interpretation of observed variability patterns, and provide simple codes for data handling.

Overall this is a well written article that describes an important dataset and merits publication in Earth System Science Data. However, I have two concerns that should be addressed before acceptance:

While horizontal aspects of the sampling strategy are well described, it is not clear to me what is the vertical configuration of the sampling strategy. The authors should describe whether the measurements were taken at a single or multiple depths, what the sampling depths are, and what is the rationale behind their selection. This is an important limitation that has to be addressed both when describing the data collection, and when discussing the observed trends

*Section 3.2. Hydrology was updated consequently. L. 155-171.*

*Information about the vertical structure of the water column has been added to the "6. Data Summary" section. L. 267-270.*

In the Discussion and Conclusion section the authors provide a review of scientific works conducted using SRN data. Although interesting by itself, to my understanding such a review is not in the scope of a data description paper, and should not be included here.

*We do not agree with this comment. As explained in the updated abstract, "We hope that this synthesis will also save data users time by allowing them to jump right into a deeper analysis based on previous conclusions and perspectives, or to investigate new scientific key challenges. These data should also be used at a wider geographical scale, combined with other data sources, to define more global patterns of environmental changes in a moving world subject to strong anthropogenic pressures. Data can also be used by the remote sensing (Ocean Color Observation) and modeling communities to calibrate or validate products in this complex and vital coastal region."*

---

## Author Response (AR2)

**29 Jan 2023**

**Topical Editor decision: Publish subject to minor revisions (review by editor)**

by Giuseppe M.R. Manzella

From the point of view of the editor, the data set is interesting and the paper is worth of publication. In future, an effort on metadata should be done. The re-use of data can be done by a larger community of scientists if an english version is provided Here a couple corrections are requested:

1) the csv files contain sampling points names. A re-use of data is much simple if the coordinates are added

2) line 86 (AL pers. comm,) Please specifify the entire name and surname. If AL is one of the authors, this must be deleted.

**13 Feb 2023**

**Author's response:**

Information available from the DOI are in English ([https://www.seanoe.org/data/00397/50832/](https://www.seanoe.org/data/00397/50832/)). We also add a supplementary csv file (S1) with a translation of every column names. As recommended, we will improve our metadata in future. We will further collaborate with Research Infrastructure such as DATA TERRA to do so.

1) The coordinates are included in the csv files available from the DOI (columns N, O, P, Q). The translation of the column names is available in S1.
2) This reference line 86 was deleted.